

# Charged eigenstate thermalization, Euclidean wormholes and global symmetries in quantum gravity

Alexandre Belin[1⋆], Jan de Boer[2†], Pranjal Nayak[3‡] and Julian Sonner[3§]

**1** CERN, Theory Division, 1 Esplanade des Particules, Genève 23, CH-1211, Suisse
**2** Institute for Theoretical Physics,University of Amsterdam,
PO Box 94485, 1090 GL Amsterdam, The Netherlands
**3** Département de Physique Théorique, Université de Genève,
24 quai Ernest-Ansermet, 1211 Genève 4, Suisse

⋆ a.belin@cern.ch   † j.deboer@uva.nl   ‡ pranjal.nayak@unige.ch   § julian.sonner@unige.ch

## Abstract

We generalize the eigenstate thermalization hypothesis to systems with global symmetries. We present two versions, one with microscopic charge conservation and one with exponentially suppressed violations. They agree for correlation functions of simple operators, but differ in the variance of charged one-point functions at finite temperature. We then apply these ideas to holography and to gravitational low-energy effective theories with a global symmetry. We show that Euclidean wormholes predict a non-zero variance for charged one-point functions, which is incompatible with microscopic charge conservation. This implies that global symmetries in quantum gravity must either be gauged or explicitly broken by non-perturbative effects.


# 1 Introduction

The thermal behavior of quantum many-body systems is well understood in terms of statistical mechanics. However, developing a microscopic understanding of thermalization is a difficult problem of sustained interest. The *eigenstate thermalization hypothesis* (ETH) [1,2] is a powerful framework to understand how a pure state can give rise to thermal behavior after sufficiently long times. The crux lies in the fact that individual eigenstates behave like a statistical ensemble for a large class of observables, with pseudo-random corrections that are exponentially small in the entropy. The ETH states that for simple (few-qubit) operators $O^a$, we have

$$\langle E_i | O^a | E_j \rangle = f^a(\bar{E})\delta_{i,j} + g^a(\bar{E}, \omega)e^{-S(\bar{E})/2}R_{ij} \,, \tag{1}$$

where $\bar{E}$ and $\omega$ are the mean energy and energy difference of the states $i$ and $j$, respectively. The matrix $R_{ij}$ is comprised of erratic order one numbers which statistically have zero mean and unit variance. In any given quantum system with fixed Hamiltonian, they are definite numbers that could be obtained by diagonalizing the Hamiltonian. However, for the purpose of computing few-point correlation functions of simple operators in high energy states, these microscopic details are irrelevant and it suffices to treat the $R_{ij}$ as true random variables. This randomness is tightly linked to the connection between quantum chaotic systems and random matrix theory (see [3] for a review).

New insights into the randomness of chaotic quantum systems have emerged from gravitational physics, through holographic duality [4]. If the chaotic quantum system at hand is a large $N$, strongly coupled conformal field theory (i.e. a *holographic* CFT), thermalization of the boundary quantum system is connected to black hole formation in the gravitational dual [5–8]. In fact, the apparent loss of unitarity in both these processes is closely related and understanding one will help in the understanding of the other. Indeed quantum thermalization has been discussed in the context of holography for precisely this reason (see for example [9–20]).

## 1.1 Randomness in Holography

It has recently become clear that the low energy effective theory on the gravity side (i.e. semiclassical general relativity and its Euclidean path integral) has the potential to know quite a lot about the structure of eigenstates of the CFT Hamiltonian, perhaps much more than we had hoped for. While it has long been known that the Bekenstein-Hawking formula computes the (coarse-grained) entropy of black hole micro-states, recent progress has established that the low energy effective theory also knows something about fine structure of the microstates and their discrete nature, for example the level-repulsion of nearby eigenvalues of the Hamiltonian [21]. New field configurations known as Euclidean wormholes contributing to the gravitational path integral play a crucial role in these developments. These may or may not be saddle-points [22–24].

Precisely quantifying the amount of CFT information that the gravitational path integral has access to has become one of the most pressing questions in holography. Interestingly, it has given a new perspective on the ETH: rather than viewing semi-classical general relativity as a traditional low energy effective theory that computes scattering amplitudes around the vacuum, it can be viewed as an effective theory in the sense of ETH, namely a framework for computing the correlators of simple operators on black hole microstates. In this context, simple operators should be understood as operators dual to supergravity fields. Multi-trace operators are also simple as long as $\Delta_O \ll N$. While there are many erratic signals in such observables that cannot be accessed through the effective theory, the moments of these signals can. This led [25] to propose a framework to describe these moments in terms of the statistics

of OPE coefficients. The OPE randomness hypothesis is a generalization of ETH that states that any index of an OPE coefficient labelling a black-hole microstate can effectively be treated as a random variable. A similar approach for Haar-typical states was studied in [26].

While ensemble-averaging over quantum systems has played a prominent role in two-dimensional gravity, for example in [21], this effective description is also applicable in individual quantum systems with a fixed Hamiltonian (at least for self-averaging quantities), which will be the focus of this work. A general framework explaining this mechanism and connecting it to random matrix theory was developed in [27] (see also [28, 29]). This framework leads to random fluctuations in OPE coefficients [30].

## 1.2 Summary of results

In this Letter, we will discuss how global symmetries interact with the ETH, wormholes and erratic signals of quantum chaos. We start by generalizing the ETH in the presence of global symmetries. For neutral operators, we can simply apply the ETH charge sector by charge sector. This is expected from a Hamiltonian decomposed into blocks corresponding to the different charge sectors, and each individual block approximates an independent random matrix [31].

Charged operators on the other hand make different charge sectors talk to one another. We discuss two possible variants of a charged ETH, one that preserves the symmetry microscopically, the other that allows for exponentially small violations of charge conservation in the random variables.[1] This second version of ETH is more relevant when viewing the ansatz as an effective theory for the simple operators, where one is agnostic about whether or not the symmetry is realized microscopically. Viewed statistically, these two ansätze give equivalent answers for low-point correlators of the simple operators in any given background. However, they differ for products of correlation functions. Most notably, we have

$$
\begin{aligned}
\langle O_q \rangle_\beta \langle O_q^\dagger \rangle_\beta \big|_{\text{c.p. ETH}} &= 0, \\
\langle O_q \rangle_\beta \langle O_q^\dagger \rangle_\beta \big|_{\text{c.v. ETH}} &\propto e^{-S},
\end{aligned}
\tag{2}
$$

where c.p. and c.v. denote the charge preserving and charge violating versions of ETH, respectively.

We will show that this resonates strongly with the gravitational perspective. In the bulk, multi-boundary Euclidean wormholes can give non-zero answers for the product of charged one-point functions. Whether the answer is non-zero depends on whether the symmetry is gauged in the bulk or not. If the symmetry is gauged, then we find a vanishing answer compatible with a charge preserving ETH, where the symmetry is realized microscopically. If on the contrary the symmetry is only a global symmetry of the bulk theory, the wormhole yields a non-zero answer. This implies that charged one-point functions have a non-zero variance and thus that OPE coefficients $C_{i\bar{i}q}$ are not exactly zero, but rather fluctuate with exponentially small variance. We show that provided that the wormhole accurately captures the variance of observables, this is inconsistent with exact global symmetries in quantum gravity. To summarize, either the symmetry is gauged in the bulk, or it is broken by non-perturbative effects in $G_N$. This provides additional evidence that global symmetries cannot exist in quantum gravity [32, 33], and extends the more recent discussion in the context of AdS/CFT using entanglement wedge reconstruction [34, 35]. Note that the violation of global symmetries due to the presence of wormholes in a gravitational theory is a long studied subject, [36–39]. The present work offers a different perspective based on ETH and quantum chaos, which is particularly relevant in light of recent developments.

---

[1]While inessential in what follows, note that these violations affect neutral operators as well.

This paper is organized as follows. In section 2, we present a charged version of the ETH. We turn to gravitational computations in section 3 and discuss correlation function on wormhole backgrounds. In section 4, we discuss the implication of our findings for global symmetries in quantum gravity. We end with a discussion of open questions in section 5.

**Note added:** while this paper was in preparation, [40,41] appeared which contain related results in the context of replica wormholes.

## 2 The ETH with global symmetries

In this section, we will present the form of the ETH which holds in the presence of global symmetries. For concreteness, we will take the global symmetry group to be $U(1)$, but it is straightforward to generalize to other groups. In the presence of a symmetry, the charge commutes with the Hamiltonian and we can simultaneously diagonalize both operators. It is thus natural to organize the Hilbert space in different charge sectors labelled by the eigenvalue $Q$ of the charge operator. Consider now a simple operator which is neutral under the global symmetry. For such operators, the generalization of the ETH is straightforward and we have

$$\langle E_i, Q_i | O^a_{q=0} | E_j, Q_j \rangle = \delta_{Q_i, Q_j}\Big( f^a(\bar{E}, Q_i)\delta_{E_i, E_j} + g^a(\bar{E}, \omega, Q_i)e^{-S(\bar{E}, Q_i)/2}R_{ij}\Big), \qquad (3)$$

where $f_a{}^2$ and $g_a$ are smooth functions of $\bar{E} \equiv (E_i + E_j)/2$, $\omega \equiv E_i - E_j$ and $Q_i$; $R_{ij}$ are random numbers with zero mean and unit variance; and, $S(\bar{E}, Q_i)$ is the microcanonical entropy in a definite charge sector $Q_i$. Note that this is just the usual form of the ETH charge sector by charge sector. An astute reader might point out that in a system with quantized energy and charge, the functions $f^a, g^a$ can't really be continuous. However, for states with macroscopic charge and energy, when the differences in these quantum numbers are much smaller than their values, we can treat them as continuous variables. This is equivalent to considering microcanonical ensembles at fixed energy and charge, and presents an immediate generalization of how these functions are defined in the conventional eigenstate thermalisation hypothesis. The above proposal, (3), is intuitively consistent with expectations from random matrix theory and quantum chaos, where one treats the different blocks of the Hamiltonian corresponding to each charge sector as independent random matrices [31] (see also [43]).

The story becomes more interesting when we discuss (simple) charged operators, since they automatically make different charge sectors talk to one another. In this case, the following ansatz should hold:

$$\begin{aligned}
\langle E_i, Q_i | O^a_q | E_j, Q_j \rangle &= \delta_{E_i, E_j}\delta_{Q_i, Q_j}\delta_{q, 0}f^a(\bar{E}, \bar{Q}) \\
&\quad + \delta_{Q_i, q+Q_j}g^a(\bar{E}, \omega, Q_i, Q_j)e^{-(S(\bar{E}, Q_i)+S(\bar{E}, Q_j))/4}R_{ij}.
\end{aligned} \qquad (4)$$

It is worthwhile to note that unlike the case of neutral operators, there is no diagonal term for operators that carry charge. This is in fact expected: the one-point function of a charged operator vanishes in the thermal (or grand-canonical) ensemble, which only leaves room for small off-diagonal contributions in the ETH ansatz. The function $g^a$ is related to the (Fourier transform of the) two-point function for the operator $O_q$, as we now show.

Let us consider the expectation value of $O^\dagger_q O_q$ in an energy eigenstate and we would like to show that this quantity has a diagonal part compatible with ETH, using only (4). To do so,

---

[2]An expression for $f$ in two-dimesional CFTs is given in [42].

we insert a resolution of the identity

$$
\begin{aligned}
\langle E_i, Q_i | O_q^\dagger O_q | E_i, Q_i \rangle &= \sum_j \langle E_i, Q_i | O_q^\dagger | E_j, Q_j \rangle \langle E_j, Q_j | O_q | E_i, Q_i \rangle \\
&= \sum_{\{|j\rangle; Q_j = Q_i + q\}} e^{-(S(E_i + \frac{\omega}{2}, Q_i) + S(E_i + \frac{\omega}{2}, Q_i + q))/2} \\
&\qquad \times |g(E_i + \frac{\omega}{2}, \omega, Q_i + q, Q_i)|^2 |R_{ij}|^2 \,.
\end{aligned}
\tag{5}
$$

The random variables $R_{ij}$ will average out to unity upon taking the sum over $j$ since they have unit variance. Moreover, we can replace the dense sum over states with varying energies, $E_j$, by an integral, namely $\sum_j \to \int d\omega \, e^{S(E_i + \omega, Q_i + q)}$, which gives

$$
\begin{aligned}
\langle E_i, Q_i | O_q^\dagger O_q | E_i, Q_i \rangle = \int d\omega \, e^{S(E_i + \omega, Q_i + q) - (S(E_i + \frac{\omega}{2}, Q_i) + S(E_i + \frac{\omega}{2}, Q_i + q))/2} \\
\times |g(E_i + \frac{\omega}{2}, \omega, Q_i + q, Q_i)|^2 \,.
\end{aligned}
\tag{6}
$$

All remaining functions are smooth and rapidly decaying functions of $\omega$ and $q$, so we can Taylor expand them to obtain to leading order

$$
\langle E_i, Q_i | O_q^\dagger O_q | E_i, Q_i \rangle \approx \int d\omega \, e^{\frac{\beta}{2}(\omega - \mu q)} |g(E_i, \omega, Q_i, Q_i)|^2 \,,
\tag{7}
$$

where we defined $\beta \equiv \frac{\partial S}{\partial E}$ and $\mu \equiv -\frac{1}{\beta} \frac{\partial S}{\partial Q}$. The result (7) should be given by the microcanonical average for the operator $O_q^\dagger O_q$ if it is to satisfy ETH, which fixes the function $g$ and its relation to the microcanonical expectation value of $O_q^\dagger O_q$.

Before moving on to discuss the implications for gravitational theories, we would like to discuss another type of charged ETH ansatz, which will mildly break charge conservation. Instead of (4), consider the ansatz

$$
\langle E_i, Q_i | O_q^a | E_j, Q_j \rangle = \delta_{E_i, E_j} \delta_{Q_i, Q_j} \delta_{q,0} f^a(\bar{E}, \bar{Q}) + \tilde{g}^a(\bar{E}, \omega, \bar{Q}, \delta Q, q) e^{-S(\bar{E}, \bar{Q})/2} R_{ij} \,.
\tag{8}
$$

The main difference between (4) and (8) is that this second version replaces the exact charge conservation by a smooth function of $\delta Q = Q_i - Q_j$ which is rapidly decaying as a function of $\delta Q - q$. From this ansatz, one could also relate the function $\tilde{g}$ to the microcanonical two-point function as in (7) (see the supplemental material for details). We would like to emphasize that the two ansätze only differ up to exponentially small corrections and are therefore indistinguishable for simple operators.

A reason to consider such a charge-breaking ansatz is the following: if we have a set of simple operators that preserve some global symmetry but we are unsure whether the microscopic Hamiltonian truly preserves this symmetry, it is perhaps more cautious to only enforce an approximate global symmetry. This would be useful for example if one wanted to formulate an effective theory for the simple operators in high energy states.

# 3   Euclidean Wormholes

In this section, we compute correlation functions of charged operators in gravitational theories. We are interested in the simplest possible setup with a wormhole solution connecting two asymptotic boundaries. The simplest solution of this type arises in AdS$_3$ when the two boundaries have negative constant curvature, hence we consider two genus-2 surfaces at the boundary.[3]

The relevant gravitational low energy effective theory is given by the Euclidean action

$$
\begin{aligned}
S &= -\frac{1}{16\pi G_N}\int d^3x\sqrt{g}\left(R+\frac{2}{\ell_{AdS}^2}\right)+S_{\text{matter}}\,,\\
S_{\text{matter}} &= \frac{1}{2}\int d^3x\sqrt{g}(|\partial\phi|^2+m^2|\phi|^2)\,.
\end{aligned}
\tag{9}
$$

Note that this action has a global $U(1)$ symmetry. The metric of this genus-2 wormhole reads

$$
ds^2=\ell_{AdS}^2(d\tau^2+\cosh^2\tau\,d\Sigma_2^2)\,,
\tag{10}
$$

where $d\Sigma^2$ is a constant curvature metric on the genus-2 surface. This geometry is locally AdS$_3$ and can be obtained from the hyperbolic ball $\mathbb{H}_3$ by taking a quotient with respect to a Fuchsian group $\Gamma$ which is a discrete subgroup of the AdS isometries.

Because the scalar theory is a free field theory and the geometry (10) is a quotient of AdS$_3$, the two-point function on the wormhole is obtained by the sum over images. For two operators inserted on opposite boundaries (see Fig. 1), the correlation function reads [22]

$$
\langle O_q\rangle_{g=2}\,\langle O_q^\dagger\rangle_{g=2}\Big|_{\text{gravity}}\sim\sum_{h\in\Gamma}\frac{1}{[\cosh(h(s))]^{2\Delta}}\,,
\tag{11}
$$

where $s$ is the distance between the 2 points on the boundary Riemann surface and $\Delta$ is the conformal dimension of the CFT operator dual to $\phi$. Here, $h\in\Gamma$ is an element of the Fuchsian discrete subgroup of the hyperbolic symmetry group, $SL(2,R)$. Correspondingly, the sum over $h(s)$ denotes the sum over all images generated under the action of this subgroup on the geodesic.[4] For sufficiently large $\Delta$, this sum converges which is related to the wormhole being (perturbatively) stable. The fact that this correlation function doesn't vanish has an interpretation in terms of the variance of the genus-2 one-point function of the operator $O$, which carries global charge. As we will discuss in the next section, this has drastic consequences for global symmetries in quantum gravity.

Before moving on to the consequences of such non-vanishing correlation functions for quantum gravity, we will first discuss the situation where the $U(1)$ global symmetry of the field $\phi$ is gauged. In this case, the boundary-to-boundary correlation function in the bulk is not gauge-invariant unless the two operators are connected by a Wilson line that propagates through the wormhole. This is depicted on the right hand side of Fig. 1. In this case, the correlation function vanishes, as already noted in [22].

The simplest way to see this is to note that in the presence of multiple boundaries, the asymptotic symmetry due to the bulk gauge field becomes one copy of the global symmetry per disconnected Euclidean boundary. In the case of our genus-2 wormhole, the boundary

---

[3]Note that because we are considering genus-2 boundaries, we are not computing thermal one-point functions and their variance but rather genus-2 one-point functions. Instead of probing the variance of $C_{\bar{i}iq}$, we instead probe $C_{\bar{l}qk}C_{ijl}\overline{C_{ijk}}$. This does not affect our conclusion for global symmetries.

[4]In the above expression we aren't carefully keeping track of the overall normalisation factor of the correlation function. which are not critical for our discussion and therefore we use '$\sim$' to remind us of this fact.

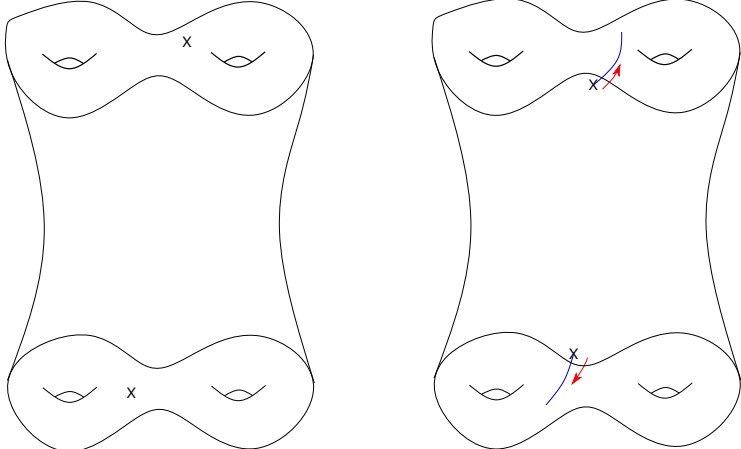

Figure 1: A genus-2 wormhole on which we compute correlation functions. On the left, the situation where the symmetry in the bulk is not gauged. This yields a non-zero correlation function. On the right, the situation where the symmetry is gauged in the bulk. In this case the field theory operators can be interpreted, using the extrapolate AdS/CFT dictionary, as the boundary limit of the bulk operator insertions which are attached to Wilson lines that end on the respective boundary. This correlation function vanishes.

global symmetry is $U(1) \times U(1)$ and the correlation function $\langle O_q \rangle_{g=2} \langle O_q^\dagger \rangle_{g=2}$ is charged under it (even if it is neutral under the diagonal subgroup). It must hence vanish. We therefore have[5]

$$\langle O_q \rangle_{g=2} \langle O_q^\dagger \rangle_{g=2} \Big|_{\text{gauged}} = 0 \,. \tag{12}$$

As demonstrated in Figure 1 the corresponding bulk object one needs to compute is a pair of insertions, $\phi(x)^\dagger W(x, \infty_{\text{top}})$ and $W(\infty_{\text{bottom}}, y)\phi(y)$, of charged operators attached to Wilson lines extending to the respective boundaries.[6] It is worthwhile to note that this does not imply that the correlation function $\langle \phi(x)^\dagger W(x, y)\phi(y) \rangle$, corresponding to the insertion of charged operators, $\phi^\dagger, \phi$, connected by a Wilson line, $W(x, y)$, vanishes for bulk points. Such correlation functions are perfectly fine gauge-invariant objects of the bulk theory, even if they are separated in Euclidean time (in particular, they are neutral under the $U(1) \times U(1)$ global symmetry of the boundary).

It is worthwhile mentioning a peculiarity of the Euclidean setup. Clearly, one can have a non-zero correlation function $\langle \phi(x)^\dagger W(x, y)\phi(y) \rangle$ when $x$ and $y$ lie on the same time-slice. In a Euclidean setup however, one can also move one of the operators in time, as long as there is a Euclidean ball surrounding both charges. In such a setup, the correlation function again is non-zero. There is therefore an interesting limit where we connect the two operators by a Wilson line and send the operators to the boundary, which now is no longer forced to vanish. We return to this question in the discussion section.

---

[5]The same argument applies to other types of symmetries, even spacetime ones. For example, a finite temperature one-point function is independent of Euclidean time. Therefore $\langle O(\tau_1) \rangle_\beta \langle O(\tau_2) \rangle_\beta$ must be independent of $\tau_1$ and $\tau_2$. A given wormhole solution may naively look like it gives a non-trivial $\tau_1 - \tau_2$ dependence, but this will be destroyed by integrating over a family of wormhole solutions that restore time translation symmetry on both boundaries. In our situation, this integral is over the gauge field.

[6]The left and the right boundaries are symbolically denoted here by $\infty_{L,R}$ here.

# 4   No Global Symmetries in Quantum Gravity

We will now show that the existence of multi-boundary Euclidean wormholes prevents the existence of exact global symmetries in quantum gravity. First, we need to discuss the difference between an exact global symmetry of quantum gravity and one that is gauged. From the CFT standpoint, both involve a graded operator algebra and exact selection rules for correlation functions: a correlation function is non-zero only if the sum of the charges of all operators vanishes. In particular, this applies to all OPE coefficient which must satisfy charge conservation

$$C_{O_{q_1} O_{q_2} O_{q_3}} \propto \delta_{q_1+q_2+q_3,0} \,. \tag{13}$$

The difference between a situation where the symmetry is gauged in the bulk involves the dual of the bulk gauge field: a CFT current which implements the action of the global symmetry.

It is currently unknown whether a consistent CFT with a local stress-tensor can have an exact symmetry for all its correlation functions without having a current (see [34] for a detailed discussion on such issues).[7] Here, we will show that the low-energy gravitational effective theory is smart enough to know that exact global symmetries are not allowed. Note that independently of whether the symmetry is gauged in the bulk, the one-point function of a charged operator must vanish on any compact Euclidean surface. In particular, we have

$$\langle O_q \rangle_{g=2} = 0 \,. \tag{14}$$

This is an exact statement, and follows from the selection rule (13). We will now see that this is in conflict with the wormhole answer.

The existence of wormhole solutions in the bulk and the correlations they imply may seem puzzling at first sight, but such wormhole correlation functions have been interpreted as encoding the variance of microscopic CFT observables as the result of some coarse-graining. For one-point functions, we have

$$\langle O \rangle \langle O \rangle \Big|_{\text{gravity}} = \overline{|\langle O \rangle_{\text{CFT}}|^2} \,, \tag{15}$$

where the $\overline{\cdot}$ notation refers to some coarse-graining involving averaging over an energy band which we will not aim to make precise here. In general, when quantities like OPE coefficients or spectral phases are erratically varying, wormhole contributions can give us the mean and variance of such signals. The central assumption that we will make is that the wormhole contribution in gravity accurately captures the CFT variance for this type of signal.

Hence, we seem to arrive at a contradiction, *reductio ad impossibile*. The wormhole correlation function (11) is non-vanishing, implying that the charged one-point function has some variance. But this is in direct contradiction with (14) which asserts that charged one-point functions are exactly zero. What this shows us is that if we try to enforce an exact global symmetry in quantum gravity, the existence of Euclidean wormholes tells us that this symmetry cannot be exact, and must necessarily be broken by non-perturbative effects. It is remarkable that the low-energy gravitational effective theory and its Euclidean path integral are smart enough to know this. On the other hand, if the symmetry is gauged in the bulk, the exact microscopic selection rule is enforced by the gauge symmetry and the variance (i.e. wormhole contribution) exactly vanishes.

Let us now make contact with the ETH ansatz in the presence of a global symmetry. We proposed two ansätze for ETH, (2), one that exactly preserves the global symmetry and one that only approximately does. Recall that for one-boundary correlators, charge conservation was

---

[7]In the absence of a local stress-tensor, such CFTs clearly exist: the canonical example is generalized free fields. In fact any QFT in AdS with a global symmetry will generate such a CFT.

not violated by large amounts for either ansatz and the two agreed up to exponentially small corrections. We conclude that the two different ansätze correspond to gravitational theories in which global symmetry is either gauged or broken by some non-perturbative phenomenon, respectively. We also learn that the multi-boundary correlation functions are the apt observables that can distinguish these effects.

# 5  Discussion

In this paper, we have presented the generalization of the ETH when there are additional global symmetries, and discussed two possible version of the ansatz: one that manifestly preserves the symmetry microscopically, and another that only preserves it for simple operators, but allows exponentially small violations of charge conservation. We then discussed a manifestation of these two scenarios for CFTs with a holographic dual in terms of Euclidean wormholes. Assuming that Euclidean wormhole computations done with the low-energy gravitational theory accurately captures moments of certain pseudo-random signals of quantum chaos in the dual CFT, we have shown that global symmetries cannot exist in quantum gravity (at least for quantum gravity in Anti-de Sitter space).

There are two possible outcomes for the fate of a global symmetry present in the low energy effective theory: it can be explicitly broken by non-perturbative effects, and we can give a lower bound on the scale of such a breaking from the gravitational action of the wormhole, namely $e^{-\ell_{AdS}/G_N}$ [25]. This is compatible with previous findings (see for example [44]). Alternatively, it can be gauged in which case the Euclidean wormhole computation vanishes and the symmetry is exact in the microscopic CFT description. In such a case, we cannot bound the magnitude of the gauge coupling.

It is interesting to observe that the absence of global symmetries in quantum gravity is tightly connected to the fact that low-energy observers can accurately resolve charges, in contrast with energy levels which are exponentially dense. Appearance of approximate global symmetries itself is not a new phenomenon in quantum theories. It is well known that both both baryon number and lepton number are approximately conserved at low energies but these global symmetries are broken at higher energies and only the difference of baryon and lepton numbers (B-L) is a preserved symmetry. In fact, in a quantum theory beyond standard model of particle physics, B-L itself might be broken. Our work provides a holographic signature of whether such global symmetries are preserved or broken.

We conclude with some open questions. The existence of Euclidean wormholes prevents the factorization of products of CFTs on disconnected manifolds, which is inconsistent with any microscopic CFT computation. While this can sometimes originate from taking ensemble averages over microscopic theories, one would also like to understand the role of wormholes in definite unitary CFTs, and how factorization is restored. There is evidence that factorization can be restored by considering certain UV ingredients like branes, which account for "non-diagonal" elements of the quasi-random variables [45–47]. One may wonder if UV ingredients could resolve the wormhole contribution (11) and the associated tension with global symmetries in quantum gravity. The point we are making here is that in the presence of an exact global symmetry, all moments of charged one-point functions must vanish, which is not what we observe for the variance. This is irrespective of how factorization is restored.

Another avenue to consider is to understand the interplay between our results concerning energy eigenstates and typical states obtained from Haar averaging in the microcanonical window [26]. It is worth pointing out that the charge violating version of ETH is very similar to formulas one would obtain when Haar-averaging over an ensemble of states that contain several different charge sectors. It would thus naturally arise in such a context.

In this paper, we came across various type of correlation functions on wormhole background. One such correlation function is $\langle \phi(x)^\dagger W(x,y)\phi(y)\rangle$, with $x$ and $y$ arbitrary bulk points. On a Euclidean wormhole, this correlation function need not vanish for points on different time slices, and in particular we can take the operators all the way to the boundary. One may now ask what this correlation function computes, and pushing on our intuition, this must be some variance. But the variance of what? The problem is that there is no one-sided correlation function with a single operator insertion, even connected to a Wilson line. This is because the Wilson line has nowhere to go. This should connect to discussions for the TFD state, where it is known that this Wilson line is a complicated CFT operator [48,49]. Perhaps thinking about eternal traversable wormholes would help, since in Euclidean signature they look like Euclidean wormholes. We hope to return to this question in the future.

Taking a step back, we are proposing that a CFT framework that can encode all observables the low energy gravitational theory has access to, is a theory of OPE coefficients treated statistically. Gravitational computations give us access to (arbitrary) moments of the statistical distribution of microscopic data.[8] In this work, we have shown that this line of thought leads to a novel argument against global symmetries in theories of quantum gravity. It would be very interesting to connect our reasoning to the standard arguments against global symmetries coming from black hole physics. We hope to return to these questions in the future.

## Acknowledgements

We are happy to thank Monica Guica, Daniel Harlow, Nabil Iqbal, Daniel Jafferis, Lampros Lamprou, Raghu Mahajan, Kyriakos Papadodimas, Gui Pimentel, Gabor Sarosi, Edgar Shaghoulian and Sasha Zhiboedov for fruitful discussions. We would also like to acknowledge Island Hopping 2020 where many useful conversations on this project took place. JdB is supported by the European Research Council under the European Unions Seventh Framework Programme (FP7/2007-2013), ERC Grant agreement ADG 834878. This work has been partially supported by the SNF through Project Grants 200020 182513, as well as the NCCR 51NF40-141869 The Mathematics of Physics (SwissMAP).

## A   Comparing ETH ansatze

In this appendix we compare the physical predictions of the two ETH ansatze discussed in the main paper. For this purpose we look at the two point function, $\langle E_i, Q_i | O_q^\dagger O_q | E_i, Q_i \rangle$. In the main text, we have discussed how this two point function is related to the subleading component, $g$, of the ETH ansatz (4). Here we demonstrate the same for the function $\tilde{g}$ and the second ETH ansatz, (8). To do so, we insert a resolution of the identity

$$
\begin{aligned}
\langle E_i, Q_i | O_q^\dagger O_q | E_i, Q_i \rangle &= \sum_{E_j, Q_j} \langle E_i, Q_i | O_q^\dagger | E_j, Q_j \rangle \langle E_j, Q_j | O_q | E_i, Q_i \rangle \\
&= \sum_{\omega, \delta Q} e^{-S(E_i + \frac{\omega}{2}, Q_i + \frac{\delta Q}{2})} \\
&\quad \times \ |\tilde{g}^a(E_i + \frac{\omega}{2}, \omega, Q_i + \frac{\delta Q}{2}, \delta Q, q)|^2 |R_{ij}|^2 .
\end{aligned}
\tag{16}
$$

The random variables $R_{ij}$ will average out to unity upon taking the sum over $j$ since they have unit variance. Once again, we can replace the dense sum over $E_j, Q_j$ by an integral,

---

[8]An interesting challenge for this program is that the gravitational theory has access to data coming from putting CFTs on arbitrary manifolds, which in $d > 2$ does not manifestly connect to the local data of the CFT (see [50]).

namely $\sum_{\omega,\delta Q} \to \int d\omega \, d\delta Q \, e^{S(E_i+\omega,Q_i+\delta Q)}$, which gives

$$\langle E_i, Q_i | O_q^\dagger O_q | E_i, Q_i \rangle = \int d\omega \, d\delta Q \, e^{S(E_i+\omega,Q_i+\delta Q)-S(E_i+\frac{\omega}{2},Q_i+\frac{\delta Q}{2})}$$
$$\times |\tilde{g}^a(E_i + \frac{\omega}{2}, \omega, Q_i + \frac{\delta Q}{2}, \delta Q, q)|^2. \qquad (17)$$

All remaining functions are smooth and rapidly decaying functions of $\omega$ and $\delta Q$, so we can Taylor expand them to obtain to leading order

$$\langle E_i, Q_i | O_q^\dagger O_q | E_i, Q_i \rangle \approx \int d\omega d\delta Q e^{\frac{\beta}{2}(\omega-\mu\delta Q)} |\tilde{g}^a(E_i, \omega, Q_i, \delta Q, q)|^2, \qquad (18)$$

where we defined $\beta \equiv \frac{\partial S}{\partial E}$ and $\mu \equiv -\frac{1}{\beta} \frac{\partial S}{\partial Q}$. It is important to note that the function $\tilde{g}$ is really a rapidly decaying function of $\delta Q - q$, rather than just $\delta Q$. Physically, this means that the integral will be sharply peaked at $\delta Q = q$, as one would expect. For this reason, we have kept the $q$ dependence explicit in the function $\tilde{g}$.

Similarly to the charge preserving ETH, the result (7) should be given by the microcanonical average for the operator $O_q^\dagger O_q$ if it is to satisfy ETH, which fixes the function $\tilde{g}$ and its relation to the microcanonical expectation value of $O_q^\dagger O_q$. This also establishes a relation between the function $g$ that appears in equation (7) and the function $\tilde{g}$.

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
