# Peer review of "Charged Eigenstate Thermalization, Euclidean Wormholes and Global Symmetries in Quantum Gravity"

_SciPost Physics, doi:SciPost Phys. 12, 059 (2022)_

## Round 1 · Referee Report · Anonymous · 2021-9-5

Strengths

1. Well embedded into prior literature.
2. Clear motivations.
3. Clear discussions and future directions.

Weaknesses

1. Technical aspects are very brief. They assume the reader is very familiar with results in prior work, and omit various definitions.
2. The manuscript would have benefited from exploring the future directions discussed. It relies on one very simple observation about global symmetries in ETH, and its interpretation in AdS/CFT.

Report

This manuscript proposes a version of the charged ETH, and what that hypothesis implies on quantum gravity via the AdS/CFT correspondence. This leads them to arguments against global symmetries in quantum gravity.

It is a well-written manuscript, but its short format obscures some aspects of the results presented. It would benefit from expanding on certain portions to help place the results on a more firm footing.

Requested changes

Since the authors are not restricted to formatting and lengths of other journals, my requested changes are basically directed to expand the text.

1. Please change the formatting of the references. Add appropriate arxiv hyperlinks.

2. More explanations for eqn (11) are needed. What is $h(s)$? I assumed it is the action of an element of $\Gamma$ acting on geodesic length. Also, what is the symbol $\sim$ indicating here?

3. Below eqn (12), $W(x,y)$ is not defined. I assume it refers to the Wilson line of the $U(1)$ gauge field between the two points. Also, add a reference to support the claim that $\langle\phi(x)^\dagger W(x,y)\phi(y)\rangle$ does not vanish if the points don't reach the boundary.

4. I'm not sure I agree with the sentence "Here, this leads to a paradox." on page 5. It is not really a paradox: as interpreted and used as in the outcome here, the tension between eqn (11) and (14) leads to a constraint that makes predictions but it is not paradoxical. I would invite the authors to reconsider the opening of that paragraph.

5. It is strange that eqn (2) is stated as a summary of results in the introduction, but never directly referred to in the main sections. This is tied to the last paragraph of section 4 (top right column page 5) being rather vague, where I suspect quoting eqn (2) would have been appropriate. Maybe the authors can connect better the content of the summary with the main text.

6. Are there any examples of Hamiltonians, or quantum systems, that one can verify that they comply with eqn (8) explicitly? Should one think about this as a spontaneous or explicit symmetry-breaking mechanism? It is not very clear what the terms "approximate global symmetry" or "mildly break charge conservation" mean in practice.

---

## Round 1 · Referee Report · Daniel Harlow · 2021-9-6

Report

This is a nice paper on Euclidean wormholes and global symmetries in quantum gravity, as well as the generalization of the Eigenstate Thermalization Hypothesis (ETH) to the case of a $U(1)$ global symmetry. There are several points where I think improvement is needed, two major and a few more minor. First the two major points:

1) The discussion of bulk gauge symmetries around figure 1 is flawed. In particular, contrary to the text and the caption of figure 1, it is not necessary for the two operators to be connected by a Wilson line. Each can instead be connected to its respective boundary. And in fact this is implicitly assumed in the text above equation 12, since it is only if we attach them to the boundaries that we get something which transforms nontrivially under $U(1)\times U(1)$. The situation with the Wilson line connecting them as shown in figure 1 is actually the correlation function $\langle \phi(x)^\dagger W(x,y)\phi(y)\rangle$ which is discussed below equation 12, and that correlator does not need to vanish even if the operators are extrapolated to the boundaries (even after extrapolating them the operator $\phi(x)^\dagger W(x,y)\phi(y)$ is still neutral under $U(1)\times U(1)$ since the transformation of the Wilson line cancels the transformations of the scalar operators.

2) The idea that Euclidean wormholes violate global symmetries has a long history, and one which is not mentioned at all in this paper. This is a referencing error that needs to be fixed. Just to give a sampling,

https://www.sciencedirect.com/science/article/abs/pii/0550321388901095?via%3Dihub

https://www.sciencedirect.com/science/article/abs/pii/0550321389905038?via%3Dihub

https://www.sciencedirect.com/science/article/abs/pii/0550321390901498?via%3Dihub

https://arxiv.org/pdf/hep-th/9502069.pdf

In addition I have some further minor comments/questions:

3) I think the notation of equations 3-8 would be greatly improved by labeling the states as $|i\rangle$, where $i$ runs over a basis of simultaneous eigenstates of $H$ and $Q$, instead of $|E_iQ_i\rangle$. The latter makes it look like we can have states with different $E$ and $Q$ for the same $i$, which is not correct. It is especially confusing e.g. in equation 5 where a sum is written over $E_j$ and $Q_j$: this should really just be a sum over $j$!

4) Below equation (3) it is said that $f_a$ and $g_a$ are smooth functions of $Q_i$. But what does it mean to be a smooth function of a discrete variable? I sort of know what they mean, but the authors are really trying to nail down a precise version of ETH with a $U(1)$ symmetry it might be worth spelling this out more.

5) Is equation (4) supposed to hold for arbitrary $q$, or only for $q$ which is small in some sense? In particular ETH is supposed to be closed under products of operators, but that doesn't seem to be the case here due to the somewhat arbitrary choice of taking the average of the entropy at the two different charges unless we make some kind of assumption about the smallness of $q$.

6) A fair number of the references are missing arxiv/journal information, these should be checked.

---

## Round 2 · Author Response

We thank the referees for the detailed reading of our papers and for the useful comments, to which we reply below. We have replied to the referee concerns in the same order as they were raised.

---

## Round 2 · List of Changes

{\bf Comments for referee 1}
1 - We have fixed the references.

2- We have added a more detailed explanation of this formula as requested by the referee.

3- We have added a definition of $W(x,y)$ below equation (3.4), and have added several remarks in the paragraphs that follow. The simplest way to see that this Wilson line is non-zero is to remember that we are working in Euclidean signature. Consider first a theory in Lorentzian signature on a compact manifold times time. One can have correlation functions of the form $\langle O(x,t=0)W(x,y)O(y,t=0) \rangle$. These are in fact the only type of operators that make sense since on a compact manifold the total net charge must be 0. In Euclidean signature however, there is no preferred notion of spatial slice (in particular the gravity computation we are doing does not have a clear Lorentzian counterpart that we should be matching). Therefore, as long as there is a Euclidean topological sphere that can encircle the two operators and the Wilson line one should imagine that such a correlation need no longer vanish. We have added an explanation clarifying this fact.

4- We have shown that assuming both that a global symmetry exists in the bulk and that the wormhole geometry computes the variance leads to a contradiction. Therefore, something has to give. But we agree with the referee that this is not really a paradox, but indeed more like a constraint that must be satisfied by consistency. We have changed the wording of the paragraph accordingly.

5- We agree with the referee, we have given a more explicit presentation of the two possibilities and have restated equation (1.2) there. This is the last paragraph before the discussion.

6- The fact that charge conservation is violated in correlation functions to us suggests an explicit breaking of the symmetry. Such effects occur in the standard model, where for example only the difference of the baryonic and leptonic $U(1)$ is a true global symmetry. We are not aware of an explicit model in quantum mechanics where something of this nature happens, but one could consider a complex SYK model, where we add by hand one coupling affecting only 4 fermions that breaks the symmetry. We don't expect this one coupling to be very important in the IR, but it certainly explicitly breaks the symmetry for high energy states.

{\bf Comments for referee 2}
1- We thank the referee for bringing up this issue. Indeed, our discussion of the Wilson line was incorrect. The standard product of CFT one-point functions (which is charged under $U(1)\times U(1)$) does not involve the Wilson line. We have adapted the text around equation (3.4) taking this into account. We have pushed the discussion of the Wilson line that extends through the wormhole to the discussion section.

2- We thank the referee for bringing up these references and agree that they should be mentioned. We have added a sentence in the introduction to this effect.

3- We believe it is natural to label states by their relevant quantum numbers, which in this case involves both energy and charge. However, we agree with the referee that the sums were not clear enough so we have modified them accordingly to make the selection rule in charge clearer.

4- The referee brings up an interesting point, but this is not very different from the usual ETH without charge. There, the smooth functions are functions of the energy differences which are discrete numbers as well. The point is simply that the sum over states can be replaced by an integral to a good approximation. There is however an important difference between charge and energy quantum numbers, namely that the charge quantum numbers are typically much more sparsely spaced (typically $\mathcal{O}(1)$) than the energy levels. For states with macroscopic charge, one should still be able to replace the sum over charges by an integral, which is what we have in mind here. We added a comment to clarify this.

5- This question is about the notion of "simple" operators. Respectfully, we disagree with the fact that ETH should be closed under products of operators, because at some point one can product enough operators together so as to make a macroscopic operator which is no longer simple. So the notion of simple operator means $q\ll S$ (or potentially some other power of $S$).

6- We have fixed the reference issue.

Apart from the changes incorporated to address the referee comments, we have also included an appendix to discuss some of the comments following equation (2.6). This in particular demonstrates more clearly the prediction for the 2-point function of the charged operators due to the ETH ansatze that we have discussed in Section 2.

You are currently on this page

Resubmission scipost_202106_00040v2 on 6 November 2021

---

## Editorial Decision

published